# Potential Impact of FDA Flavor Enforcement Policy on Vaping Behavior on Twitter

**DOI:** 10.3390/ijerph191912836

**Published:** 2022-10-07

**Authors:** Zidian Xie, Jinlong Ruan, Yifan Jiang, Bokai Zhang, Tianlang Chen, Jiebo Luo, Dongmei Li

**Affiliations:** 1Department of Clinical & Translational Research, University of Rochester Medical Center, Rochester, NY 14642, USA; 2Department of Computer Science, University of Rochester, Rochester, NY 14627, USA

**Keywords:** electronic cigarette, FDA flavor enforcement policy, demographic, Twitter

## Abstract

In January 2020, the FDA announced an electronic cigarette (e-cigarette) flavor enforcement policy to restrict the sale of all unauthorized cartridge-based flavored e-cigarettes except tobacco and menthol flavors, which was implemented on 6 February 2020. This study aimed to understand the potential influence of this policy on one vaping behavior change—quitting vaping—using Twitter data. Twitter posts (tweets) related to e-cigarettes were collected between June 2019 and October 2020 through a Twitter streaming API. Based on the geolocation and keywords related to quitting vaping, tweets mentioning quitting vaping from the US were filtered. The demographics (age and gender) of Twitter users who mentioned quitting vaping were further inferred using a deep learning algorithm (deepFace). The proportion of tweets and Twitter users mentioning quitting vaping were compared between before and after the announcement and implementation of the flavor policy. Compared to before the FDA flavor policy, the proportion of tweets (from 0.11% to 0.20% and 0.24%) and Twitter users (from 0.15% to 0.70% and 0.86%) mentioning quitting vaping were significantly higher after the announcement and implementation of the policy (*p*-value < 0.001). In addition, there was an increasing trend in the proportion of female and young adults (18–35 years old) mentioning quitting vaping on Twitter after the announcement and implementation of the policy compared to that before the policy. Our results showed that the FDA flavor enforcement policy did have a positive impact on quitting vaping on Twitter. Our study provides an initial evaluation of the potential influence of the FDA flavor enforcement policy on user vaping behavior.

## 1. Introduction

Electronic cigarettes (e-cigarettes) are one of the tobacco products that release aerosols by heating a liquid including propylene glycol or glycerol, nicotine, and flavoring chemicals [1]. E-cigarettes have been promoted as an alternative to cigarette smoking [2]. With the aggressive marketing of e-cigarette companies and various attractive flavors, the prevalence of e-cigarette use increased significantly in middle school and high school students in recent years. Among high school students in the US, the use of e-cigarettes increased dramatically from 1.1% in 2011 to 19.6% in 2020 [3]. Among the students between 12 and 17 years old who use e-cigarettes, 82.9% of them used flavored e-cigarettes [3,4]. While there was a slight decrease in the prevalence of e-cigarette use in 2021, there were still 1.72 million high school students and 320,000 middle school students in the US who reported current e-cigarette use [5].

While long-term health effects of e-cigarette use await further investigation, recent studies showed several health effects associated with e-cigarette use, including respiratory symptoms or diseases, cardiovascular diseases, and cognitive defects [6,7,8,9,10,11,12,13,14]. The outbreak of e-cigarette, or vaping, product use associated with lung injury (EVALI) in 2019 in the United States further showed the serious health effects of e-cigarette use [15,16,17]. Therefore, it is of importance to reduce or prevent e-cigarette use in order to protect public health, especially among youth and adults who never used any tobacco products.

The appeals of various flavors have been identified as one of the major reasons for attracting youth to e-cigarette use [18,19]. In response to the epidemic of e-cigarette use, especially among youth, on 2 January 2021 the United States Food and Drug Administration (FDA) announced a flavor enforcement policy to restrict the sales of all cartridge-based unauthorized flavored e-cigarettes other than tobacco and menthol flavors. The FDA flavor enforcement policy was implemented on 6 February 2020 [20]. However, the impact of FDA flavor enforcement policy on e-cigarette use behavior remains to be evaluated.

Twitter, a popular social media platform in the United States, has been used to examine public perceptions of flavored e-cigarettes and identify topics discussed about e-cigarettes [21,22,23,24]. In addition, Twitter data have been used to examine public perceptions and discussions of government policies related to e-cigarettes, such as e-cigarette campaigns and e-cigarette flavor policies [25,26,27]. Compared to other social media platforms (such as Meta and WeChat), Twitter has many advantages. For example, Twitter data is relatively easier to access, and provides more information at the user level. Therefore, Twitter data provide a rich data source for examining the impact of the FDA flavor enforcement policy on e-cigarette use.

In this study, using Twitter data, we aimed to investigate the potential impact of the FDA flavor enforcement policy on one e-cigarette use behavior change—quitting vaping—by applying natural language processing techniques. Using a deep learning algorithm, we inferred demographic characteristics such as the age and gender of Twitter users who mentioned quitting vaping and examined their changes with the announcement and implementation of the FDA flavor enforcement policy.

## 2. Materials and Methods

### 2.1. Data Source and Preprocessing

Twitter data on e-cigarettes were collected between June 2019 and October 2020 using a Twitter streaming application programming interface (API) with e-cigarette related keywords, including “e-cig”, “e-cigs”, “ecig”, “ecigs”, “electroniccigarette”, “vape”, “vapers”, “vaping”, “vapes”, “e-liquid”, “ejuice”, “eliquid”, “e-juice”, “vapercon”, “vapeon”, “vapefam”, “vapenation”, and “juul” [28]. Furthermore, we identified tweets from the United States based on the geolocation information provided in tweets [29]. Lastly, we filtered out commercial posts related to e-cigarettes based on the promotion keywords, including “dealer”, “deal”, “customer”, “promotion”, “promo”, “promos”, “discount”, “sale”, “free shipping”, “sell”, “$”, “%”, “dollar”, “offer”, “percent off”, “store”, “save”, “price”, and “wholesale” [23,30]. After data preprocessing, we obtained 1,027,136 tweets in total. We divided the final relevant tweets into three time periods: (1) before the announcement of the FDA flavor enforcement policy (13 June 2019 to 31 July 2019), with 165,782 tweets; (2) between the announcement and the implementation of the FDA flavor enforcement policy (2 January 2020 to 5 February 2020), with 169,375 tweets; and (3) after the implementation of the FDA flavor enforcement policy (6 February 2020 to 12 October 2020), with 691,979 tweets. Due to the announcement and implementation of other local policies on flavored e-cigarettes in several states (such as New York, Michigan, New Jersey, etc.) in late 2019, we did not include Twitter data from August 2019 to December 2019 in our study to reduce the noise.

### 2.2. Tweets Related to Quitting Vaping

To filter the tweets that mentioned quitting vaping, we employed frequent itemset mining to identify the list of relevant keywords. We utilized the FP-growth (frequent pattern growth) algorithm to perform the task of frequent itemset mining. First, we substituted all punctuation with spaces, then tokenized each transaction by space. Furthermore, we removed stop words, including but not limited to “I”, “is”, “are”, “do”, “we”, “it”, “in”, “on”, “at”, “to”, and “the”. Then, we updated the list with more irrelevant words and characters by lowering the support threshold and association rule probability threshold. In the end, the support threshold was set to 1000, and the association rule probability threshold was set to 0.30. The terms “stop” and “vaping” have a support of 2037 and an association rule probability of 0.53, The terms “quit” and “vaping” have a support of 2459 and an association rule probability of 0.30. Thus, we decided to use “stop” and “quit” and their variants as keywords to identify tweets mentioning quitting vaping because they have the highest support within the dataset and the highest association with “vaping”. To compare the proportion of tweets and Twitter users mentioning quitting vaping between different periods, the two-proportion Z test was conducted with a significance level set at 5%.

### 2.3. Demographic Inference of Twitter Users

We employed deepFace, a deep learning facial recognition algorithm, to obtain an inference of the age and gender of Twitter users [14]. The deepFace API takes a user profile image as input and outputs the inferred age and gender of the user if there is one and only one face detected in the profile image. Considering the sample size, Twitter users were grouped into two age groups, 18 to 35 years old (young adults), and 35 years old and above (old adults).

## 3. Results

### 3.1. Potential Impact of FDA Flavor Enforcement Policy on One Vaping Behavior Change on Twitter

With the announcement and implementation of the FDA flavor enforcement policy, it was important to evaluate how this policy affected one of the anticipated e-cigarette use behavior changes: quitting vaping. In total, we identified 2178 tweets mentioning quitting vaping, such as “i quit smokin the juul 1 week ago today”, “Well just got through the last of my vape juice so I’m officially quitting”, and “Looks like I quit vaping just in time”. As shown in Figure 1, the proportion of tweets mentioning quitting vaping was 0.20% (345/169,375) between the announcement and implementation of the FDA flavor enforcement policy (2 January 2020–5 February 2020), and 0.24% (1644/691,979) after the implementation of this policy (6 February 2020–12 October 2020), which was significantly higher than 0.11% (189/165,782) before the announcement of this policy (13 June 2019–31 July 2019), with both having a *p*-value < 0.001. Furthermore, we compared the number of unique Twitter users who mentioned quitting vaping between different periods (Figure 1). Relative to before the announcement of the FDA flavor enforcement policy (185/126,735 = 0.15%), the proportion of Twitter users who mentioned quitting vaping after the announcement (340/48,368 = 0.70%) and implementation of this policy (1624/189,686 = 0.86%) was significantly higher, with both having a *p*-value < 0.001.

### 3.2. Demographic Characteristics of Twitter Users Who Mentioned Quitting Vaping

While we observed that the proportion of Twitter users who mentioned quitting vaping significantly increased with the announcement and implementation of the FDA flavor enforcement policy, it is very important to understand the demographic characteristics of Twitter users who mentioned quitting vaping with the announcement and implementation of the FDA flavor enforcement policy. Among 2149 unique Twitter users who mentioned quitting vaping in our study, 220 of them (10.24%) could be inferred in terms of their age and gender. As shown in Figure 2, compared to before the announcement of the FDA flavor enforcement policy (7/20 = 35.00%), the percentage of females among Twitter users who mentioned quitting vaping was slightly higher after the announcement (16/39 = 41.03%) and implementation (65/161 = 40.37%) of this policy, even though it was not statistically significant (*p*-value > 0.05). In addition, as shown in Figure 3, the proportion of users within the age group 18–35 among Twitter users who mentioned quitting vaping was slightly higher after the announcement (35/39 = 89.74%) and implementation (139/161 = 86.34%) of FDA flavor enforcement policy compared to that (14/20 = 70.00%) before this policy, even though it also was not statistically significant.

## 4. Discussion

In this study, using Twitter data, we showed that the proportion of US Twitter users mentioning quitting vaping significantly increased after the announcement and implementation of the FDA flavor enforcement policy. Furthermore, we showed that females and young adults were slightly more likely to mention quitting vaping on Twitter after the announcement and implementation of the FDA flavor enforcement policy.

The objective of any tobacco product policy or ban is to reduce or prevent the use of targeted tobacco products. One previous study showed that with the FDA menthol cigarette ban in 2009, among US tobacco users about 25–64% would attempt to quit smoking and 11–46% would consider switching to other tobacco products [31]. Another study showed that the predicted probability of smoking among youth and young adults was reduced from 43% to 27% with the national ban on flavored cigarette products [32]. In this study, with the FDA flavor enforcement policy, more Twitter users mentioned quitting vaping on Twitter, which is similar to previous findings with the FDA flavored cigarette ban. Although it awaits further investigation, it might be reasonable to speculate that vapers might consider quitting vaping due to the unavailability of their favorite e-cigarette flavors. Among Twitter users who mentioned quitting vaping, the proportion of females and young adults slightly increased after the announcement and implementation of the FDA flavor enforcement policy. One previous study based on a survey among San Francisco residents identified the decreasing usage of flavored tobacco products among young people after the local flavor ban [33]. Therefore, the FDA flavor enforcement policy might have more impact on females and young adults.

In this study, we showed there was a significant increase in mentioning quitting vaping on Twitter after the announcement and implementation of the FDA flavor enforcement policy, suggesting that the FDA flavor enforcement policy had some positive effects on one user behavior change: quitting vaping. While more studies with a large sample size are needed to further validate our findings, our study showed some preliminary but valuable evidence about the potential impacts of the FDA flavor enforcement policy on one user vaping behavior change, which provides useful guidance for future tobacco regulatory policymaking.

There were several limitations in this study. First, only Twitter data were used to investigate the influence of the FDA flavor enforcement policy, whereas Twitter users cannot represent the general population. In addition, Twitter is not the most popular social media platform among young people. Therefore, our findings might not apply to adolescents. Based on a previous study on Twitter users’ geographical locations [34], people living in urban areas are more likely to use Twitter, leading to an under-representation of people living in rural areas. Second, due to the lack of demographic information provided directly by Twitter users, we estimated the demographic information (age and gender) based on Twitter users’ profile images. However, some users do not provide valid photos, and the estimation of demographics may not be very accurate due to the limitation of the face-recognition algorithm. Therefore, the sample size was very small in this study, which might introduce some biases and lead to insignificant differences between different gender and age groups. Third, the main purpose of the FDA flavor enforcement policy is to prevent e-cigarette use among youth. While this study only examined the potential effects of this flavor policy on adults, further study on youth is urgently needed. Fourth, in this study, we did not filter out or control the co-use of other illicit drugs with e-cigarettes, which might be very difficult to quit vaping. Fifth, we did not eliminate the situation of quitting vaping due to any clinical morbidity or pregnancy status. However, we do not expect a significant change in the prevalence of chronic diseases and pregnancy after the FDA flavor enforcement policy. Lastly, with the COVID-19 pandemic in the United States starting from March 2020, it is technically challenging to remove some potential influence of the COVID-19 pandemic on e-cigarette use, such as “I actually quit vaping since all of this corona stuff”.

## 5. Conclusions

Our results provided valuable information about the potential influence of the FDA flavor enforcement policy on quitting e-cigarette use on Twitter, which provides valuable feedback on the FDA flavor enforcement policy.

## Figures and Tables

**Figure 1 ijerph-19-12836-f001:**
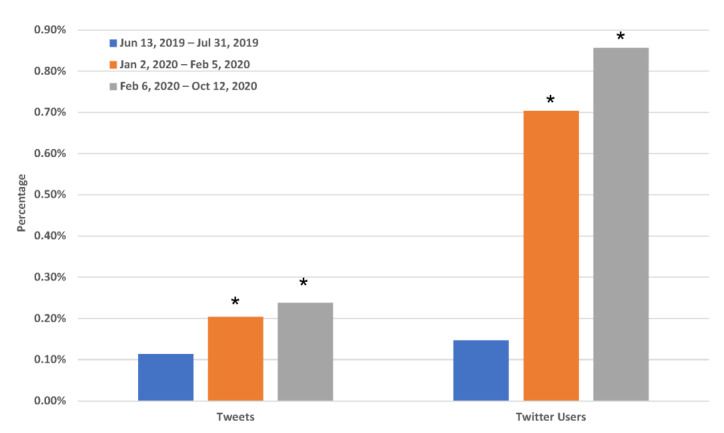
Changes in the proportion of tweets and Twitter users mentioning quitting vaping after the announcement and implementation of the FDA flavor enforcement policy. 13 June 2019–31 July 2019: before the announcement of the FDA flavor enforcement policy; 2 January 2020–5 February 2020: between the announcement and the implementation of the FDA flavor enforcement policy; 6 February 2020–12 October 2020: after the implementation of the FDA flavor enforcement policy. *: *p*-value < 0.001.

**Figure 2 ijerph-19-12836-f002:**
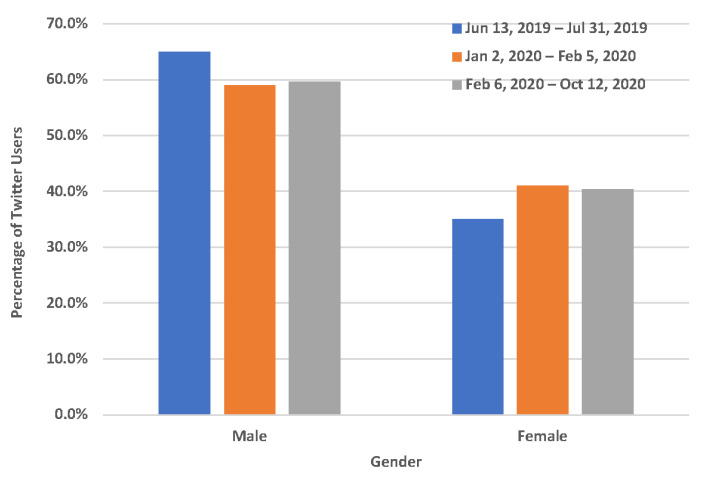
Gender characteristics of Twitter users who mentioned quitting vaping. 13 June 2019–31 July 2019: before the announcement of the FDA flavor enforcement policy; 2 January 2020–5 February 2020: between the announcement and the implementation of the FDA flavor enforcement policy; 6 February 2020–12 October 2020: after the implementation of the FDA flavor enforcement policy.

**Figure 3 ijerph-19-12836-f003:**
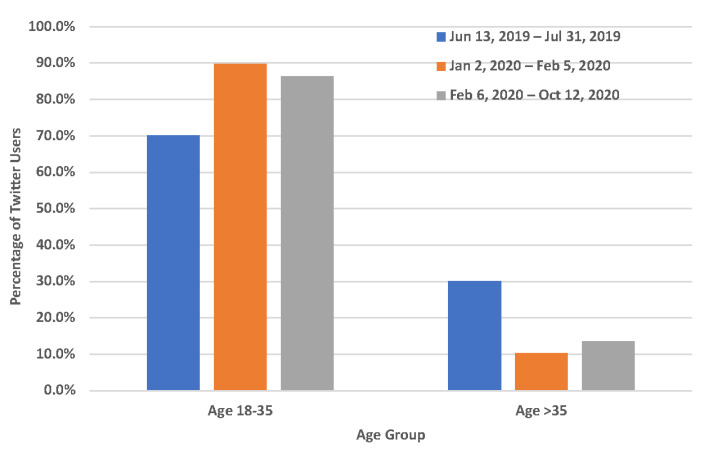
Age characteristics of Twitter users who mentioned quitting vaping. 13 June 2019–31 July 2019: before the announcement of the FDA flavor enforcement policy; 2 January 2020–5 February 2020: between the announcement and the implementation of the FDA flavor enforcement policy; 6 February 2020–12 October 2020: after the implementation of the FDA flavor enforcement policy.

## Data Availability

The data and scripts used for analysis and creating figures are available on request from the corresponding author, DL.

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
