# Peer review of "Potential Impact of FDA Flavor Enforcement Policy on Vaping Behavior on Twitter"

_ijerph, 2022, doi:10.3390/ijerph191912836_

Round 1

Reviewer 1 Report

This manuscript provides important findings around the impact of FDA flavour enforcement policy on user vaping behaviour. This article will contribute to the literature on how quitting vaping significantly increased among US tweeter users especially females and young adults after the announcement and implementation of the FDA flavour enforcement policy. Overall, this is a very interesting, timely and strong paper; I just have a few minor comments for clarity and consistency.

Although the rationale for use of the Twitter platform has been given, however, it is worth mentioning why other social media platforms like Meta, Reddit, WeChat and so on were not considered.

How nicotine containing e-cigarettes or the use of e-cigarettes with other illicit drugs were filtered or controlled or eliminated in this study, such measures were missing. It is worth mentioning because nicotine or illicit drug (often mixed with different favours such as mango, strawberry, or mint) containing vaping products are very difficult to quit and evidence shows that such quitting of e-cigarettes (with a combination of nicotine/illicit products) requires special professional assistance.

The author mentioned the limitations of demographic data, and additionally, how quitting vaping with any clinical morbidity or pregnancy status (as findings include female quitting significantly increased) or chronic diseases were eliminated during the screening process of data collection.

Author Response

Overview of responses to reviewers’ comments

Manuscript ID: ijerph-1939325

We really appreciate the critical comments and suggestions from the editor and three reviewers. We have revised our manuscript based on reviewers’ comments. The details are listed below point by point.

This manuscript provides important findings around the impact of FDA flavour enforcement policy on user vaping behaviour. This article will contribute to the literature on how quitting vaping significantly increased among US tweeter users especially females and young adults after the announcement and implementation of the FDA flavour enforcement policy. Overall, this is a very interesting, timely and strong paper; I just have a few minor comments for clarity and consistency.

Response: Thanks so much for all positive comments!

Although the rationale for use of the Twitter platform has been given, however, it is worth mentioning why other social media platforms like Meta, Reddit, WeChat and so on were not considered.

Response: Thanks so much for the suggestions. We have added the reason for not using other social media platforms in the Introduction section of our revised manuscript.

“Compared to other social media platforms (such as Meta and WeChat), Twitter has many advantages. For example, Twitter data is relatively easier to access, and provides us more information at the user level.”

How nicotine containing e-cigarettes or the use of e-cigarettes with other illicit drugs were filtered or controlled or eliminated in this study, such measures were missing. It is worth mentioning because nicotine or illicit drug (often mixed with different favours such as mango, strawberry, or mint) containing vaping products are very difficult to quit and evidence shows that such quitting of e-cigarettes (with a combination of nicotine/illicit products) requires special professional assistance.

Response: We appreciate the reviewer’s comments. We used e-cigarette-related tweets without filtering out or controlling for the possible co-use of other illicit drugs. We have put this as a limitation in our revised manuscript.

“Fourth, in this study we did not filter out or control the co-use of other illicit drugs with e-cigarettes, which might be very difficult to quit.”

The author mentioned the limitations of demographic data, and additionally, how quitting vaping with any clinical morbidity or pregnancy status (as findings include female quitting significantly increased) or chronic diseases were eliminated during the screening process of data collection.

Response: We appreciate the reviewer’s comments. We did not eliminate quitting vaping due to clinical morbidity or pregnancy status. We have added this as a limitation in our revised manuscript.

“Fifth, we did not eliminate the situation of quitting vaping due to any clinical morbidity or pregnancy status. However, we do not expect a significant change in the prevalence of chronic diseases and pregnancy after the FDA flavor enforcement policy.”

Reviewer 2 Report

This paper analize the potencial impact of FDA flavor enforcement policy and vaping behavior in the social network of Twitter. This paper in novelty and relevant for the topic of vaping behavior. This behavior is todavy relevant in adolescents and adults. For this, the FDA made a policy restriction of the menthol flavors, that it´s utilized by the industry to increase the consumptions of adolescents and young adults. 

This study is centered in the social network of Twitter, that it´s utilized by young, and fundamentally by adults. The selection is correct, because the study consider two groups: young adults (18-35 years) and old adults (35 + years), comparing in both the pre- and post-FDA flavor enforcement policy. The results of the study confirm the efficacy of this policy, with a increase of quitting vaping.

However, the author need made a change in p. 4, parragraph 2, because not is correct indicat that in Figure 3 quitting vaping was greater, because it was not statistically significant. Please, check this, and all the results of the paper about this. Only it´s possible speack about decrease if is a statistical significant difference.

The discussion is balanced.

In limitations of the study the author need indicate that Twitter not is the social network more utilized by the young people. 

Author Response

Overview of responses to reviewers’ comments

Manuscript ID: ijerph-1939325

 We really appreciate the critical comments and suggestions from the editor and three reviewers. We have revised our manuscript based on reviewers’ comments. The details are listed below point by point.

This paper analize the potencial impact of FDA flavor enforcement policy and vaping behavior in the social network of Twitter. This paper in novelty and relevant for the topic of vaping behavior. This behavior is todavy relevant in adolescents and adults. For this, the FDA made a policy restriction of the menthol flavors, that it´s utilized by the industry to increase the consumptions of adolescents and young adults.

Response: Thanks so much for the positive comments.

This study is centered in the social network of Twitter, that it´s utilized by young, and fundamentally by adults. The selection is correct, because the study consider two groups: young adults (18-35 years) and old adults (35 + years), comparing in both the pre- and post-FDA flavor enforcement policy. The results of the study confirm the efficacy of this policy, with a increase of quitting vaping.

Response: Thanks so much for the comments.

However, the author need made a change in p. 4, parragraph 2, because not is correct indicat that in Figure 3 quitting vaping was greater, because it was not statistically significant. Please, check this, and all the results of the paper about this. Only it´s possible speack about decrease if is a statistical significant difference.

Response: Thanks for the comments. We agree with the reviewer that the differences between genders and age groups were not statistically significant, although the point estimator was higher. Therefore, instead saying “higher”, we changed it into “slightly higher” in Page 4 Paragraph 2. We did mention the difference was not statistically significant. In addition, in the abstract we revised “there is an increasing trend” to consider the insignificant differences. The statistical insignificance might be due to the relatively small sample size. We also added it to the limitation section of our revised manuscript.

“In addition, there is an increasing trend in the proportion of female and young adults (18-35 years old) mentioning quitting vaping on Twitter after the announcement and implementation of the policy compared to that before the policy.”

“Therefore, the sample size was very small in this study, which might introduce some biases and lead to insignificant differences between different gender and age groups.”

The discussion is balanced.

Response: Thanks so much for the comments.

In limitations of the study the author need indicate that Twitter not is the social network more utilized by the young people. 

Response: We appreciate the reviewer’s comments. We have added this into the limitation section of our revised manuscript.

“In addition, Twitter is not the most popular social media platform among young people. Therefore, our findings might not apply to the adolescents.”